# Noninvasive Mapping of Extracellular Potassium in Breast Tumors via Multi-Wavelength Photoacoustic Imaging

**DOI:** 10.3390/s25154724

**Published:** 2025-07-31

**Authors:** Jeff Folz, Ahmad Eido, Maria E. Gonzalez, Roberta Caruso, Xueding Wang, Celina G. Kleer, Janggun Jo

**Affiliations:** 1Department of Biomedical Engineering, University of Michigan, Ann Arbor, MI 48109, USA; 2Department of Pathology, University of Michigan Medical School, Ann Arbor, MI 48109, USA; 3Rogel Cancer Center, University of Michigan Medical School, Ann Arbor, MI 48109, USA

**Keywords:** photoacoustic, chemical imaging, breast cancer, CCN6, metaplastic breast carcinoma, nanoparticle, potassium

## Abstract

Elevated extracellular potassium (K^+^) in the tumor microenvironment (TME) of breast and other cancers is increasingly recognized as a critical factor influencing tumor progression and immune suppression. Current methods for noninvasive mapping of the potassium distribution in tumors are limited. Here, we employed photoacoustic chemical imaging (PACI) with a solvatochromic dye-based, potassium-sensitive nanoprobe (SDKNP) to quantitatively visualize extracellular potassium levels in an orthotopic metaplastic breast cancer mouse model, Ccn6-KO. Tumors of three distinct sizes (5 mm, 10 mm, and 20 mm) were imaged using multi-wavelength photoacoustic imaging at five laser wavelengths (560, 576, 584, 605, and 625 nm). Potassium concentration maps derived from spectral unmixing of the photoacoustic images at the five laser wavelengths revealed significantly increased potassium levels in larger tumors, confirmed independently by inductively coupled plasma mass spectrometry (ICP-MS). The PACI results matched ICP-MS measurements, validating PACI as a robust, noninvasive imaging modality for potassium mapping in tumors *in vivo*. This work establishes PACI as a promising tool for studying the chemical properties of the TME and provides a foundation for future studies evaluating the immunotherapy response through ionic biomarker imaging.

## 1. Introduction

The tumor microenvironment (TME) plays a central role in cancer progression, resistance to therapy, and immune system evasion in breast and other cancers. A defining feature of the TME is its abnormal chemical composition, including hypoxia (reduced oxygen concentration), acidosis (decreased pH), and hyperkalemia (elevated extracellular potassium, K^+^) [1,2,3,4]. These conditions frequently co-exist in solid tumors and contribute to treatment failure by creating a physiologically hostile and therapy-resistant environment. Among them, hyperkalemia, i.e., an increased K^+^ concentration, has recently gained attention as a potentially important but relatively under-investigated factor in tumor biology.

K^+^ is the most abundant intracellular cation. Under normal physiological conditions, the intracellular K^+^ concentration exceeds 100 mM [5,6,7], whereas the extracellular level is maintained around 5 mM. In solid tumors, especially within necrotic regions that develop under hypoxic and nutrient-deprived conditions, cellular lysis results in the release of intracellular K^+^ into the extracellular space. This process can increase the local extracellular K^+^ concentration by 5- to 10-fold [6,8,9]. Such elevated K^+^ levels have been shown to suppress T-cell effector functions and impair immune surveillance [8,10], indicating that K^+^ accumulation in the TME may serve as a mechanism by which tumors evade the host immune response.

Although the significance of extracellular K^+^ in regulating the TME is increasingly recognized, current imaging techniques are not well-suited to measuring K^+^ levels *in vivo* with a sufficient spatial and temporal resolution. ^39^K magnetic resonance imaging (MRI) [11,12] has been used for K^+^ mapping but is limited by a low sensitivity, poor spatial resolution, and high operational cost. Fluorescence-based sensors offer high sensitivity in vitro but are unsuitable for deep-tissue imaging due to strong optical scattering in biological tissues [13,14,15]. Ion-selective microelectrodes provide accurate measurements but are invasive and lack the capability to generate spatial maps of K^+^ distribution in intact tissues. These limitations highlight the need for a new imaging approach that can noninvasively quantify extracellular K^+^
*in vivo* with a high resolution.

To overcome this gap, we developed a photoacoustic chemical imaging (PACI) technique [13,16,17,18,19,20,21] using an ionophore-based, potassium-sensitive nanoprobe. This solvatochromic dye-based K^+^-sensing nanoparticle (SDKNP) undergoes a change in optical absorption in response to K^+^ binding, which can be detected through PACI [22]. PACI combines the molecular contrast of optical techniques with the high spatial resolution and deep tissue penetration of ultrasound, enabling real-time, noninvasive imaging of the tissue chemical composition *in vivo*.

We have previously demonstrated the application of PACI for quantitative *in vivo* imaging of extracellular K^+^ concentrations in tumor tissue using a subcutaneous glioma mouse model [22]. By injecting SDKNP into subcutaneous tumor models in mice, we capture high-resolution K^+^ maps and validate them through inductively coupled plasma mass spectrometry (ICP-MS) [8]. We further examine how K^+^ levels vary with tumor size, uncovering insights into the evolution of hyperkalemia in tumor progression.

This work establishes PACI as a powerful tool for mapping functional biomarkers in the TME and opens up new approaches for investigating the chemical underpinnings of cancer therapy resistance. In the long term, K^+^ imaging may provide critical guidance for personalized treatment strategies and deepen our understanding of the biochemical landscape of solid tumors.

## 2. Materials and Methods

### 2.1. Chemicals

Millipore water was used for all syntheses and buffer preparations. Chemicals purchased from Sigma-Aldrich (St. Louis, MS, USA) include Pluronic F-127, valinomycin, bis(2-ethylhexyl) sebacate, sodium tetrakis[3,5-bis(trifluoromethyl)phenyl]borate, potassium chloride, acetonitrile, acetic anhydride, 1-iodooctadecane, 2-methylbenzothiazole, (dimethylamino)cinnamaldehyde, sodium iodide, diethyl ether, and methanol.

### 2.2. Synthesis of SDKNP

Synthesis of SDKNP was first reported by Eric Bakker’s group, and few changes were made to the original protocol [23].

#### 2.2.1. 2-methyl-3-octadecylbenzo[d]thiazol-3-ium (SD2a)

2-methylbenzothiazole (1.5 g, 10 mmol) and 1-iodooctadecane (3.8 g, 10 mmol) were dissolved and refluxed in acetonitrile for 24 h. The solution was removed from heat and allowed to solidify. The crude product was further precipitated in diethyl ether, collected, and washed several times in diethyl ether. Once dried, the product was used without any further purification.

#### 2.2.2. 2-((1E,3E)-4-(4-(dimethylamino)phenyl)buta-1,3-dien-1-yl)-3-octadecylbenzo[d]thiazol-3-ium iodide (SD2)

SD2a (265 mg, 0.7 mmol) along with (dimethylamino)cinnamaldehyde (122 mg, 0.7 mmol) was dissolved in acetic anhydride and refluxed for 20 min. The reaction solution was then poured into a warm solution of 10 mM sodium iodide (in Millipore water). The dark purple precipitate was washed several times with water, dried, collected, and used without any further purification.

#### 2.2.3. SDKNP

SD2 (0.2 mg, 0.3 μmol), sodium tetrakis[3,5-bis(trifluoromethyl)phenyl]borate (0.9 mg, 1 μmol), valinomycin (1.2 mg, 1 μmol), Pluronic F-127 (5 mg, 0.4 μmol), and bis (2-ethylhexyl) sebacate (8 mg, 19 μmol, 8.75 μL) were dissolved in 3 mL of methanol. The methanol cocktail was then injected into 30 mL of Millipore water under vigorous stirring. The surface of the methanol–water mixture was blasted with argon gas for 1 h to remove the methanol. The nanoparticle solution was concentrated to the desired concentration using an Amicon Ultra-15 centrifuge filter (100 k Da). The schematic diagram of SDKNP is shown in Appendix A.

### 2.3. Animal Models

All animal procedures were approved by the University Committee on the Use and Care of Animals (UCUCA) at the University of Michigan (Protocol #PRO00011264; PI: Janggun Jo). Mice were housed in the Unit for Laboratory Animal Medicine (ULAM) at the University of Michigan Medical School. Orthotopic xenograft mouse models of breast cancer were used in this study. Eight-week-old female FVB/NJ mice (The Jackson Laboratory, stock #001800) were orthotopically injected with 5 × 10^5^ breast cancer cells derived from a mouse model of metaplastic breast carcinoma generated by our lab based on mammary specific conditional Ccn6 knockout (Ccn6-KO) suspended in 50 μL of Matrigel into the right inguinal mammary fat pad [24,25,26,27,28,29,30,31]. Tumor growth was monitored daily using caliper measurements. Mice were selected for imaging when tumors reached target sizes of 0.5 cm, 1.0 cm, or 2.0 cm in diameter. At these defined tumor sizes, animals were used for both photoacoustic K^+^ imaging and quantitative analysis of tumor K^+^ concentration by inductively coupled plasma mass spectrometry (ICP-MS) [32], which served as the reference standard.

### 2.4. Multi-Wavelength Photoacoustic Ratiometric Imaging of K^+^

To image extracellular K^+^ levels *in vivo* using SDKNP, we employed a multi-wavelength photoacoustic ratiometric imaging technique previously developed by our group [13,22]. Photoacoustic imaging was conducted at five laser excitation wavelengths: 560, 576, 584, 605, and 625 nm. These wavelengths were selected based on isosbestic points and regions of maximal spectral separation among different concentrations of SDKNP, oxy-hemoglobin, and deoxy-hemoglobin (Hb/HbO_2_) to optimize the spectral unmixing process (Figure 1). Specifically, 560 nm corresponds to the isosbestic point of SDKNP, 576 nm to the absorption peak of HbO_2_, 584 nm to the isosbestic point of hemoglobins, and 605 nm to the absorption peak of SDKNP. To minimize errors from wavelength-dependent optical attenuation, we intentionally selected five excitation wavelengths within a narrow spectral range (560–625 nm), where the tissue fluence distribution is relatively uniform. This design balances spectral separation of SDKNP, Hb, and HbO_2_ with minimal variation in light penetration, enhancing the reliability of multi-wavelength spectral unmixing. A tunable optical parametric oscillator (OPO) laser system, pumped by the third harmonic (355 nm) of a Nd:YAG laser (Surelite, Amplitude, Milpitas, CA, USA), was used for excitation, delivering 5 ns pulses at a repetition rate of 10 Hz [13,22]. The applied optical energy density on the skin surface was approximately 15 mJ/cm^2^, which is well below the 20 mJ/cm^2^ safety delimited by the American National Standard Institute (ANSI) at the selected wavelengths. The photoacoustic and ultrasound dual-modality imaging system consisted of a programmable research ultrasound platform (Vantage 256, Verasonics, Kirkland, WA, USA) integrated with a 128-element linear-array transducer (CL15-7, Philips, Andover, MA, USA) with a center frequency range of 7–15 MHz. The system allows for the sequential collection of 50 data points for both PA and US images, with each modality being acquired for about 5 s. For photoacoustic imaging, mice bearing orthotopic Ccn6-KO tumors of different sizes (0.5, 1.0, and 2.0 cm) were locally injected with 50 µL of 10 mg/mL SDKNP into the tumor. Photoacoustic signals were acquired at each of the five wavelengths, with 50-frame signal averaging performed per wavelength to enhance the signal-to-noise ratio. Spectral unmixing and ratiometric analysis were performed on a pixel-by-pixel basis using custom MATLAB scripts (R2022b, MathWorks, Natick, MA, USA). The method is shown in the Appendix A. From the five-wavelength image sets, the photoacoustic intensities of each pixel were introduced to separate the contributions to spectroscopic photoacoustic measurement from the two forms of hemoglobin and the SDKNP. For the multi-wavelength photoacoustic ratiometric imaging of potassium, we used the optical absorption changes of SDKNP depending on the K^+^ concentration (Figure 2A) and the ratios relative to the photoacoustic intensity at 560 nm (Figure 2B). As the previous studies for the PACI technique [13,22], K^+^ concentration maps were generated from the breast tumor models, enabling quantitative evaluation of the extracellular K^+^ distribution in tumors of varying sizes.

### 2.5. Collection of Tumor Interstitial Fluid

Tumor interstitial fluid samples were collected using previously established protocols [32]. Briefly, surgically resected tumors had their surfaces rinsed with 0.9% saline solution before being gently blotted dry to remove surface moisture and blood. Tumors were then placed in Eppendorf tubes and centrifuged at 106 g for 10 min. Obtained tumor interstitial fluid volumes were typically between 1 and 5 μL.

### 2.6. Inductively Coupled Plasma Mass Spectrometry

Tumor interstitial fluid samples were diluted by a factor of 100,000 in Millipore water before being filtered using a 0.2 μm syringe filter. Samples were analyzed using a Perkin Elmer Nexion 2000 ICP-MS instrument (Perkin Elmer, Shelton, CT, USA).

## 3. Results

To investigate the relationship between tumor size and extracellular K^+^ levels, we employed PACI using a K^+^-sensitive nanoprobe, SDKNP. The probe was developed in the Bakker group [23], and its characterization is summarized in Figure 3. Due to the employment of the potassium ionophore valinomycin, the probe displays excellent selectivity over the sodium ion, and minimal interference from calcium at physiological concentrations (Figure 3A). The radius of the nanosensors is approximately 50 nm on average, and no significant change in size is observed between high- and low-potassium environments (Figure 3B). An average zeta potential of −30 mV indicates that the probe is stable and unlikely to flocculate (Figure 3C).

Mice bearing orthotopic Ccn6-KO metaplastic breast cancer tumors were divided into three groups based on tumor size, as measured along the longest dimension: Group 1 (5 mm, *n* = 4), Group 2 (10 mm, *n* = 3), and Group 3 (20 mm, *n* = 4). After the intratumor injection of SDKNP, multi-wavelength photoacoustic imaging was performed at five selected wavelengths (560, 576, 584, 605, and 625 nm). K^+^ distribution maps were generated using spectral unmixing and ratiometric analysis (Figure 4A). Given that the 20 mm tumors were positioned with a significant portion extending above the abdominal skin, directly under the mammary gland pad, roughly 10 mm of laser penetration was adequate to cover the majority of the tumor volume.

In Figure 4A, representative K^+^ maps from each group show the spatial distribution of K^+^ in the tumors. Visual inspection suggests an increase in K^+^ signal intensity with tumor size. A quantitative analysis of mean K^+^ levels across all animals within each group is presented in Figure 4B. Group-wise comparisons using two-sample *t*-tests revealed a statistically significant difference between Group 1 and Group 3 (*p* = 0.0059), indicating the presence of elevated K^+^ levels in larger tumors (Figure 4B). Comparisons between Group 1 and Group 2 (*p* = 0.1413) and between Group 2 and Group 3 (*p* = 0.0535) did not reach statistical significance, although the latter showed a trend toward significance.

Following photoacoustic K^+^ imaging, tumors were collected, and the extracellular K^+^ content was quantified using ICP-MS. The results are shown in Figure 5, which presents the group-wise averages and standard deviations for K^+^ levels. The ICP-MS results were in strong agreement with the PACI-derived K^+^ levels derived from background-subtracted signals. Similar trends were observed, with Group 3 exhibiting the highest K^+^ levels among the three groups. Two-sample *t*-tests confirmed statistically significant differences between Group 1 and Group 3 (*p* = 0.0113), while no significant differences were observed between Group 1 and Group 2 or Group 2 and Group 3. These results collectively demonstrate that tumor size is positively correlated with extracellular K^+^ accumulation and validate the PACI-based K^+^ imaging approach as a reliable noninvasive method for assessing the TME ionic composition *in vivo*.

## 4. Discussion

In this study, we successfully applied PACI with a K^+^-sensitive nanoprobe (SDKNP) to noninvasively visualize extracellular K^+^ levels *in vivo* in orthotopic Ccn6-KO mataplastic breast carcinomas of varying sizes. These constitute the most aggressive form of triple negative breast cancer with limited effective therapeutic options and uarded outcome. Our results demonstrate that the K^+^ concentration in the TME increases with the tumor size, as shown both by PACI and independent validation using ICP-MS. Elevated extracellular K^+^ has previously been implicated in immune suppression within the TME, acting as a byproduct of necrotic cell death and contributing to tumor resistance against immunotherapy [24,31,33,34]. Our data support the hypothesis that larger, more advanced tumors contain higher levels of extracellular K^+^, potentially reflecting an increased necrotic burden and metabolic stress, which warrants further investigation. 

A statistically significant difference in K^+^ levels was observed between the smallest (5 mm) and largest (20 mm) tumors, with both PACI and ICP-MS showing consistent results. This validates the effectiveness of PACI as a reliable tool for *in vivo*, noninvasive K^+^ mapping, offering a promising alternative to invasive techniques such as tumor fluid extraction and electrode-based measurements. Since the selected wavelengths for ratiometric imaging fall within a narrow spectral range (560–625 nm), the light fluence within tissue is expected to be similar across wavelengths, reducing the impact of depth-dependent attenuation on quantification accuracy. Therefore, no additional fluence correction was applied in this study. Separate from the technical considerations of the imaging system, in this study, the sample size for each group was relatively small, which may have limited our ability to detect statistically significant differences between adjacent and smaller tumor sizes (e.g., 10 mm vs. 20 mm). This study was designed as a proof-of-concept demonstration of noninvasive extracellular K^+^ imaging using PACI. Despite the modest group sizes (*n* = 3–4 per group), both PACI and ICP-MS showed consistent trends, and significant differences were detected between the smallest and largest tumors. Nevertheless, we acknowledge that larger sample sizes in future studies will be necessary to enhance the statistical power and confirm subtle differences between intermediate tumor stages. While a trend toward increased K^+^ was observed, the *p*-value between Group 2 and Group 3 was very close to 5%, but did not reach significance, possibly due to insufficient statistical power. Future studies with larger animal cohorts could be warranted to confirm these findings and to better characterize the dynamic relationship between tumor growth and ionic changes in the TME.

Additionally, this work represents a significant step toward functional molecular imaging of tumor biochemistry using aggressive breast cancer as a model. PACI-based K^+^ imaging has potential applications not only in monitoring tumor progression but also in predicting the response to therapy and evaluating the efficacy of immuno-oncological interventions. The findings presented in this study lay an important foundation for our ongoing research on immunotherapy’s efficacy.

## 5. Conclusions

This study demonstrates that photoacoustic chemical imaging using a potassium-sensitive nanoprobe can effectively map extracellular potassium levels in aggressive triple negative breast cancer tumors of varying sizes. Photoacoustic imaging results showed good agreement with inductively coupled plasma mass spectrometry used for validation, confirming the method’s accuracy and reliability. Importantly, the tumor size was positively correlated with potassium levels, supporting the hypothesis that potassium accumulation reflects tumor progression and may influence therapeutic outcomes. Future work should explore longitudinal imaging, correlation with histopathology and immune markers, and integration with multi-parametric photoacoustic chemical imaging (e.g., simultaneous pH and oxygen imaging) to provide a comprehensive view of the tumor microenvironment. With further development, photoacoustic chemical imaging has the potential to become a valuable tool in personalized oncology by enabling noninvasive monitoring of chemical tumor biomarkers *in vivo*.

## Figures and Tables

**Figure 1 sensors-25-04724-f001:**
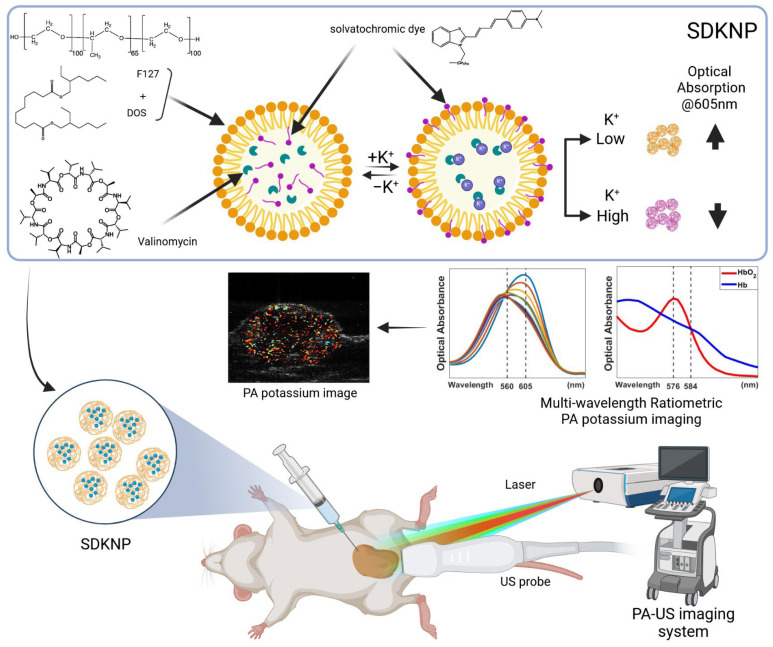
Illustration of photoacoustic K^+^ imaging with solvatochromic dye-based K^+^-sensing nanoparticles (SDKNP). The SDKNP potassium probe has a characteristic absorption change depending on the K^+^ levels. The K^+^ image was obtained using a multi-wavelength ratiometric photoacoustic imaging method.

**Figure 2 sensors-25-04724-f002:**
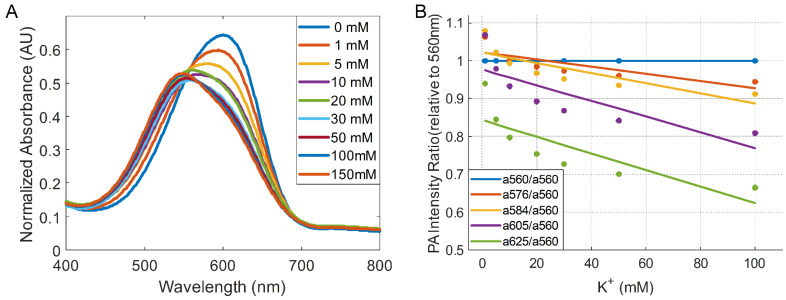
Optical properties of SDKNP. (**A**) Spectroscopic optical absorption of SDKNP at different K^+^ levels. (**B**) Absorption ratios between each wavelength relative to 560 nm, which is the isosbestic point of the absorption changes.

**Figure 3 sensors-25-04724-f003:**
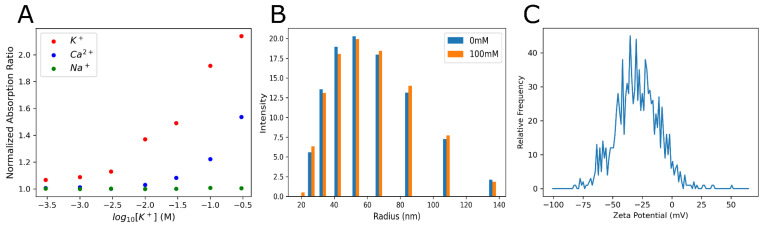
Calibration and characterization of SDKNP. (**A**) A ratiometric calibration of SDKNPs’ optical absorption changes against various ions. Ratios of the absorption peaks at 520 nm over 600 nm were plotted against ion concentrations. All measurements were taken at a constant ionic strength using LiCl to maintain constant tonicity. (**B**) Dynamic light scattering measurements of SDKNP size at both 0 and 100 mM K^+^, again using LiCl to maintain constant tonicity. (**C**) Measurement and distribution of zeta potential for the SDKNPs.

**Figure 4 sensors-25-04724-f004:**
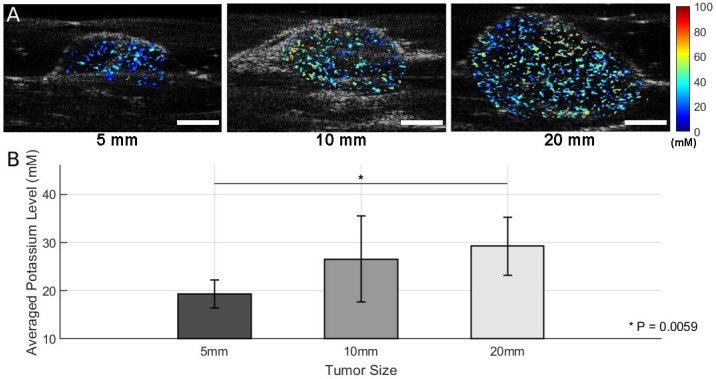
Results of photoacoustic K^+^ imaging with SDKNP. (**A**) Representative K^+^ maps acquired by the photoacoustic and ultrasound dual-modality imaging system. Color-coded K^+^ concentration maps were superimposed on grayscale ultrasound images for anatomical localization. (**B**) K^+^ levels in tumors of three different size groups (Group 1 [5 mm]: *n* = 4; Group 2 [10 mm]: *n* = 3; Group 3 [20 mm]: *n* = 4) Group-wise mean potassium levels ± standard deviations are shown. Statistical comparisons were performed using two-sample *t*-tests. * *p* < 0.01. Scale bar = 5 mm.

**Figure 5 sensors-25-04724-f005:**
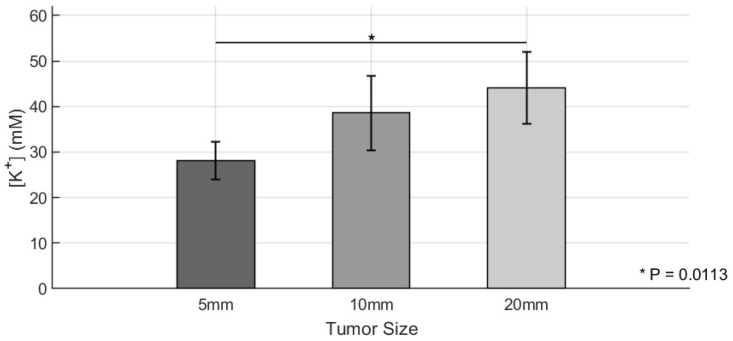
Quantitative analysis of Ccn6-KO tumor K^+^ levels using ICP-MS. K^+^ levels in tumors of three different size groups (Group 1 [5 mm]: *n* = 4; Group 2 [10 mm]: *n* = 3; Group 3 [20 mm]: *n* = 4) were quantified using inductively coupled plasma mass spectrometry (ICP-MS). Bar graphs show mean values ± standard deviations for potassium levels calculated by background-subtracted signals in the three groups. Statistical comparisons were conducted using two-sample *t*-tests. * *p* < 0.05.

## Data Availability

The data that support the finding of this study are available from the corresponding authors on reasonable request.

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
