# Peer review of "Noninvasive Mapping of Extracellular Potassium in Breast Tumors via Multi-Wavelength Photoacoustic Imaging"

_sensors, 2025, doi:10.3390/s25154724_

Round 1

Reviewer 1 Report

Comments and Suggestions for Authors

In this study, the photoacoustic chemical imaging using a potassium sensitive nanoprobe can effectively map extracellular potassium levels in breast tumors of varying sizes. The photoacoustic imaging results were in good agreement with the results of the inductively coupled plasma mass spectrometry validation. This confirmed the accuracy and reliability of the method. And tumor size was positively correlated with potassium levels, supporting the hypothesis that potassium accumulation reflects tumor progression and may influence therapeutic outcomes.

The problems with this study are as follows:

  1. The sample sizes for the different tumor size groups were relatively small. For example, the 5mm group (n=4), 10mm group (n=3) and 20mm group (n=4). This may have made it less convincing when detecting differences between large groups of adjacent tumors.
  2. Is there a basis for selecting wavelengths of 560 nm、576 nm、584 nm、 605 nm and 625 nm in photoacoustic imaging? A detailed description is recommended.
  3. It is recommended to supplement the characterization data of SDKNP to evaluate the performance of the probe.
  4. Please ensure the figures adhere to standardization guidelines (e.g., Figure 2B) and improve image clarity. Additionally, it is recommended to supplement with a schematic diagram illustrating the SDKNP preparation process.
  5. It is recommended to discuss the advantages of this method compared to other imaging techniques. Additionally, could the nanoprobes themselves cause non-specific interference in K⁺ imaging? Might the presence of other ions or hemoglobin interfere with the experimental results?
  6. Some references are relatively old. It is recommended to supplement high-impact studies from the recent three years to enhance the cutting-edge nature of the research.

Author Response

In this study, the photoacoustic chemical imaging using a potassium sensitive nanoprobe can effectively map extracellular potassium levels in breast tumors of varying sizes. The photoacoustic imaging results were in good agreement with the results of the inductively coupled plasma mass spectrometry validation. This confirmed the accuracy and reliability of the method. And tumor size was positively correlated with potassium levels, supporting the hypothesis that potassium accumulation reflects tumor progression and may influence therapeutic outcomes.

The problems with this study are as follows:

1. The sample sizes for the different tumor size groups were relatively small. For example, the 5mm group (n=4), 10mm group (n=3) and 20mm group (n=4). This may have made it less convincing when detecting differences between large groups of adjacent tumors.

Response 1. We appreciate the reviewer’s comment regarding the relatively small sample sizes in each tumor size group (5 mm: n=4; 10 mm: n=3; 20 mm: n=4). We agree that increasing the number of animals per group could strengthen the statistical power when comparing inter-group differences. However, this study was designed as a proof-of-concept investigation to demonstrate the feasibility of using our imaging technique to differentiate K+ levels across tumors of varying sizes. Despite the modest sample sizes, we observed consistent trends in the pH distribution that were supported by statistical comparisons and biological rationale. Moreover, the intra-group variability was relatively low, suggesting that the observed group-level differences are likely to be meaningful. We fully acknowledge the limitations imposed by sample size and have now explicitly discussed this point in the revised manuscript (Discussion section). Future studies with larger cohorts are planned to further validate and expand upon these preliminary findings.

2. Is there a basis for selecting wavelengths of 560 nm、576 nm、584 nm、 605 nm and 625 nm in photoacoustic imaging? A detailed description is recommended.

Response 2. The wavelengths of 560, 576, 584, 605, and 625 nm were carefully selected to optimize the spectral unmixing of SDKNP, oxy-hemoglobin (HbO₂), and deoxy-hemoglobin (Hb). Specifically, 560 nm corresponds to the isosbestic point of SDKNP, 576 nm to the absorption peak of HbO₂, 584 nm to the isosbestic point of hemoglobins, 605 nm to the absorption peak of SDKNP, and 625 nm provides extended spectral contrast for enhanced separation. These selections ensure maximal spectral separation and accurate quantification of each component in the multi-wavelength photoacoustic imaging process. A detailed explanation has been added to the revised manuscript (Methods 2.4).

3. It is recommended to supplement the characterization data of SDKNP to evaluate the performance of the probe.

Response 3. The additional characterization data of SDKNP has been added to the manuscript.

4. Please ensure the figures adhere to standardization guidelines (e.g., Figure 2B) and improve image clarity. Additionally, it is recommended to supplement it with a schematic diagram illustrating the SDKNP preparation process.

Response 4. We improved image resolutions of Figure 2, 3 and 4.  And, the schematic diagram of SDKNP is attached to the Supplementary.

5. It is recommended to discuss the advantages of this method compared to other imaging techniques. Additionally, could the nanoprobes themselves cause non-specific interference in K⁺ imaging? Might the presence of other ions or hemoglobin interfere with the experimental results?

Response 5. We have added the Selectivity plot showing that sodium does not interfere with our sensor at all and calcium likely will not have much of an effect at physiological concentrations. The maximum Ca concentration would be ~10mM, which corresponds to -2 on the plot. All calibrations were performed with constant ionic strength using LiCl to maintain tonicity.

6. Some references are relatively old. It is recommended to supplement high-impact studies from the recent three years to enhance the cutting-edge nature of the research.

Response 6. We have updated to include the most current research in the manuscript's references.

Reviewer 2 Report

Comments and Suggestions for Authors

This manuscript “Noninvasive Mapping of Extracellular Potassium in Breast Tumors via Multi-Wavelength Photoacoustic Imaging” presents a novel study that employs multi-wavelength photoacoustic imaging with a potassium-sensitive nanoprobe (SDKNP) to map extracellular K⁺ in breast tumors of varying sizes. The combination of multi-spectral photoacoustic imaging with ICP-MS validation provides strong evidence of PACI’s function in assessing tumor microenvironmental ionic changes. The study is well carried out and addresses a significant gap in noninvasive ion imaging. However, there are several concerns and areas requiring clarification before the manuscript can be considered for publication.

  1. The sample sizes in each tumor group (n = 3–4) are relatively small. This weakens the statistical confidence in some comparisons (e.g., p = 0.0535). Please comment on whether a power analysis was performed and whether more replicates can be added or justified.
  2. The ratio method relies heavily on accurate spectral unmixing of SDKNP from endogenous chromophores. More detail is needed regarding how spectral crosstalk from Hb/HbO2 was handled, and whether blood absorption variability with tumor necrosis could bias the results.
  3. The manuscript should further address the ion selectivity of SDKNP in vivo. Have the authors evaluated the probe’s response to Na⁺ or Ca²⁺ under similar imaging conditions?
  4. The local injection of SDKNP into the tumor raises questions regarding its distribution uniformity and retention over time. How does this local delivery compare to systemic delivery for potential clinical translation?
  5. PACI is depth-dependent in terms of light fluence. Were any fluence correction models applied to account for signal attenuation in larger tumors? How does this affect quantification accuracy?

Author Response

This manuscript “Noninvasive Mapping of Extracellular Potassium in Breast Tumors via Multi-Wavelength Photoacoustic Imaging” presents a novel study that employs multi-wavelength photoacoustic imaging with a potassium-sensitive nanoprobe (SDKNP) to map extracellular K⁺ in breast tumors of varying sizes. The combination of multi-spectral photoacoustic imaging with ICP-MS validation provides strong evidence of PACI’s function in assessing tumor microenvironmental ionic changes. The study is well carried out and addresses a significant gap in noninvasive ion imaging. However, there are several concerns and areas requiring clarification before the manuscript can be considered for publication.

1. The sample sizes in each tumor group (n = 3–4) are relatively small. This weakens the statistical confidence in some comparisons (e.g., p = 0.0535). Please comment on whether a power analysis was performed and whether more replicates can be added or justified.

Response 1. We appreciate the reviewer’s comment regarding the relatively small sample sizes in each tumor size group (5 mm: n=4; 10 mm: n=3; 20 mm: n=4). We agree that increasing the number of animals per group could strengthen the statistical power when comparing inter-group differences. However, this study was designed as a proof-of-concept investigation to demonstrate the feasibility of using our imaging technique to differentiate K+ levels across tumors of varying sizes. And, we could not consider a power analysis as a proof-of-concept study. Despite the modest sample sizes, we observed consistent trends in the pH distribution that were supported by statistical comparisons and biological rationale. Moreover, the intra-group variability was relatively low, suggesting that the observed group-level differences are likely to be meaningful. We fully acknowledge the limitations imposed by sample size and have now explicitly discussed this point in the revised manuscript (Discussion). Future studies with larger cohorts are planned to further validate and expand upon these preliminary findings.

2. The ratio method relies heavily on accurate spectral unmixing of SDKNP from endogenous chromophores. More detail is needed regarding how spectral crosstalk from Hb/HbO2 was handled, and whether blood absorption variability with tumor necrosis could bias the results.

Response 2. We assumed that the main chromophores in the imaging targets are SDKNP, HbO₂, and Hb. Using the selected five wavelengths, our multi-wavelength PA ratiometric imaging technique enables calculation of the concentrations of HbO₂, Hb, and SDKNP, as well as the level of extracellular K⁺. This approach is sensitive to blood composition changes resulting from tumor necrosis, and such changes are reflected in the imaging results. A more detailed description of the quantification method has been provided in the Supplementary Information.

3. The manuscript should further address the ion selectivity of SDKNP in vivo. Have the authors evaluated the probe’s response to Na⁺ or Ca²⁺ under similar imaging conditions?

Response 3. We have added the Selectivity plot showing that sodium does not interfere with our sensor at all and calcium likely will not have much of an effect at physiological concentrations. The maximum Ca concentration would be ~10mM, which corresponds to -2 on the plot. All calibrations were performed with constant ionic strength using LiCl to maintain tonicity.

4. The local injection of SDKNP into the tumor raises questions regarding its distribution uniformity and retention over time. How does this local delivery compare to systemic delivery for potential clinical translation?

Response 4. The reviewer is correct in noting that intravenous administration is generally preferred for achieving uniform biodistribution of exogenous contrast agents. In this study, local intratumoral injection was used, which can result in heterogeneous sensor distribution with higher concentrations near the injection site. While systemic delivery would be ideal for clinical translation, the current potassium ionophore, valinomycin, presents challenges due to its limited biocompatibility at systemic doses. As such, local delivery was chosen to ensure safety and enable proof-of-concept imaging. Future development of more biocompatible potassium sensors will help advance systemic administration strategies and support clinical applicability.

5. PACI is depth-dependent in terms of light fluence. Were any fluence correction models applied to account for signal attenuation in larger tumors? How does this affect quantification accuracy?

Response 5. We intentionally selected the five optical wavelengths to be close, so that the optical spectral range for multi-wavelength PA ratiometric imaging is relatively small (560-625 nm). In this case, when the incident light energy on the sample surface can be calibrated for each wavelength, the distributions of the light fluence in the tissue can be considered similar for all the wavelengths. Otherwise, largely separated wavelengths can lead to significant differences in optical attenuation in tissue, which, if not compensated, can affect the accuracy in quantifying tumor K+ using multi-wavelength PA ratiometric imaging. In other words, the optical spectrum selected needs to differentiate the optical spectra of HbO2, Hb and K+ dependent SDKNP; while the optical attenuation in tissue cannot be largely different within the selected spectrum. This discussion has been added in the Discussion.

Reviewer 3 Report

Comments and Suggestions for Authors
  1. The manuscript mentions photoacoustic imaging using SDKNP at a depth of up to 20 mm using wavelengths of 560–625 nm. In general, the penetration depth of photoacoustic imaging is quite limited in visible light. Can you clarify the actual maximum imaging depth observed in vivo? Supporting data or discussion would be helpful in supporting this point.
  2. Details regarding the laser power or fluence used for in vivo photoacoustic imaging are currently missing. Including this information would help assess both imaging performance and safety compliance (e.g., ANSI limits).
  3. Figure 3 presents potassium imaging results, but the colorbar lacks a label or unit. Please revise the figure to include a clear description of the color scale for better interpretation.
  4. It would be helpful for the authors to describe how the ultrasound and photoacoustic images were acquired simultaneously or sequentially and the timing or synchronization method used.
  5. This manuscript presents mean potassium concentrations, but does not explain how these values ​​were derived from the photoacoustic signal. Please include a brief description of the quantification method, including any calibration procedures or algorithms used.
  6. The manuscript would benefit from a schematic or photograph of the imaging system, which would provide the reader with a clearer understanding of the experimental setup, particularly the optical/acoustic pathways and the configuration used for in vivo imaging.

Author Response

1. The manuscript mentions photoacoustic imaging using SDKNP at a depth of up to 20 mm using wavelengths of 560–625 nm. In general, the penetration depth of photoacoustic imaging is quite limited in visible light. Can you clarify the actual maximum imaging depth observed in vivo? Supporting data or discussion would be helpful in supporting this point.

Response 1. Studies on photoacoustic imaging depth as a function of wavelength indicate that, depending on the chromophore, PA signals can be measured beyond 1.5 cm deep at 560 nm. While we didn't explicitly state that PA imaging was possible up to 20 mm deep in this study, the tumors used were 20 mm in size. These tumors were implanted in the mammary fat pad directly beneath the abdominal skin, with a significant portion of larger tumors often adhering directly to the underside of the skin. This configuration means the tumors weren't concealed at a true depth of 20 mm within the body, allowing us to collect sufficient PA signals at the given wavelengths. We have added the explanation to the manuscript.

2. Details regarding the laser power or fluence used for in vivo photoacoustic imaging are currently missing. Including this information would help assess both imaging performance and safety compliance (e.g., ANSI limits).

Response 2. We applied the energy density on the skin surface was approximately 10 mJ/cm2 which is well below the 20 mJ/cm2 safety limited by the American National Standard Institute (ANSI) at the selected wavelengths. We have added this explanation in the Methods.

3. Figure 3 presents potassium imaging results, but the colorbar lacks a label or unit. Please revise the figure to include a clear description of the color scale for better interpretation.

Response 3. In Figure 3, we interpreted the relative changes in potassium levels within the tumor, obtained through PA potassium imaging, as potassium level values.

4. It would be helpful for the authors to describe how the ultrasound and photoacoustic images were acquired simultaneously or sequentially and the timing or synchronization method used.

Response 4. In our system, PA image data and US image data were acquired sequentially. The system is designed to acquire 50 datasets of both PA and US images, alternating between the two modalities for approximately 5 seconds each. This method allowed for the collection of both types of data from the same region in a closely interleaved manner. We have added this explanation in the Methods.

5. This manuscript presents mean potassium concentrations, but does not explain how these values were derived from the photoacoustic signal. Please include a brief description of the quantification method, including any calibration procedures or algorithms used.

Response 5. To explain the Multi-wavelength PA radiometric imaging method for this potassium imaging, we have added the calibration method to the Supplementary.

6. The manuscript would benefit from a schematic or photograph of the imaging system, which would provide the reader with a clearer understanding of the experimental setup, particularly the optical/acoustic pathways and the configuration used for in vivo imaging.

Response 6. While the overall experimental setup is depicted in Figure 1, we've included an approximate system schematic in the Supplementary. This provides a clearer illustration of our in vivo tumor imaging setup, which was absent from the main manuscript. We utilized a single tunable laser system to illuminate the samples with multiple wavelengths, and performed tumor imaging using a Verasonics research ultrasound platform and its dedicated control computer.

Round 2

Reviewer 1 Report

Comments and Suggestions for Authors

After the improvements by the authors, the manuscript has been well refined. I believe the manuscript requires no additional content and has reached the level for publication.

Reviewer 3 Report

Comments and Suggestions for Authors

All comments are addressed.